# Clinical and Microbiological Impact of Implementing a Decision Support Algorithm through Microbiologic Rapid Diagnosis in Critically Ill Patients: An Epidemiological Retrospective Pre-/Post-Intervention Study

**DOI:** 10.3390/biomedicines11123330

**Published:** 2023-12-16

**Authors:** Alejandro Rodríguez, Frederic Gómez, Carolina Sarvisé, Cristina Gutiérrez, Montserrat Galofre Giralt, María Dolores Guerrero-Torres, Sergio Pardo-Granell, Ester Picó-Plana, Clara Benavent-Bofill, Sandra Trefler, Julen Berrueta, Laura Canadell, Laura Claverias, Erika Esteve Pitarch, Montserrat Olona, Graciano García Pardo, Xavier Teixidó, Laura Bordonado, María Teresa Sans, María Bodí

**Affiliations:** 1Critical Care Department, Hospital Universitari de Tarragona Joan XXIII, Mallafre Guasch 4, 43005 Tarragona, Spain; sitrefler@yahoo.es (S.T.); julen.berrueta@estudiants.urv.cat (J.B.); lauraclaverias@gmail.com (L.C.); mbodi.hj23.ics@gencat.cat (M.B.); 2Faculty of Medicine, Department of Basic Medical Sciences, Rovira & Virgili University, 43005 Tarragona, Spain; lcanadell.hj23.ics@gencat.cat; 3Pere Virgili Health Research Institute (IISPV), 43005 Tarragona, Spain; ffgomez.hj23.ics@gencat.cat (F.G.); csarvise.tgn.ics@gencat.cat (C.S.); cgutierrez.hj23.ics@gencat.cat (C.G.); mgiralt.hj23.ics@gencat.cat (M.G.G.); maguerrero.hj23.ics@gencat.cat (M.D.G.-T.); separdo.hj23.ics@gencat.cat (S.P.-G.); eplana.hj23.ics@gencat.cat (E.P.-P.); cbenavent.hj23.ics@gencat.cat (C.B.-B.); eepitarch.hj23.ics@gencat.cat (E.E.P.); molona.hj23.ics@gencat.cat (M.O.); ggarciap.hj23.ics@gencat.cat (G.G.P.); tsans.ebre.ics@gencat.cat (M.T.S.); 4Centre for Biomedical Research in Respiratory Diseases Network (CIBERES), 43005 Tarragona, Spain; 5Microbiology/Clinical Analysis Laboratory, Hospital Universitari de Tarragona Joan XXIII, 43005 Tarragona, Spain; 6Faculty of Medicine, Department of Medicine and Surgery, Rovira & Virgili University, 43005 Tarragona, Spain; 7Centre for Biomedical Research in Infectious Diseases Network (CIBERINFEC), 28220 Madrid, Spain; 8Molecular Biology/Clinical Analysis Laboratory, Hospital Universitari de Tarragona Joan XXIII, 43005 Tarragona, Spain; 9Tarragona Health Data Research Working Group (THeDaR), Critical Care Department, Hospital Universitari de Tarragona Joan XXIII, 43005 Tarragona, Spain; 10Hospital Pharmacy, Hospital Universitari de Tarragona Joan XXIII, 43005 Tarragona, Spain; 11Preventive Medicine, Infection Control Group, Hospital Universitari de Tarragona Joan XXIII, 43005 Tarragona, Spain; 12ICU Nursing, Hospital Universitari de Tarragona Joan XXIII, 43005 Tarragona, Spain; xteixido.hj23.ics@gencat.cat (X.T.); lauraborpe@gmail.com (L.B.)

**Keywords:** nosocomial infection, rapid microbiological diagnostic techniques, antimicrobial consumption, defined daily dose, ventilator-associated pneumonia, FilmArrays

## Abstract

Background: Data on the benefits of rapid microbiological testing on antimicrobial consumption (AC) and antimicrobial resistance patterns (ARPs) are scarce. We evaluated the impact of a protocol based on rapid techniques on AC and ARP in intensive care (ICU) patients. Methods: A retrospective pre- (2018) and post-intervention (2019–2021) study was conducted in ICU patients. A rapid diagnostic algorithm was applied starting in 2019 in patients with a lower respiratory tract infection. The incidence of nosocomial infections, ARPs, and AC as DDDs (defined daily doses) were monitored. Results: A total of 3635 patients were included: 987 in the pre-intervention group and 2648 in the post-intervention group. The median age was 60 years, the sample was 64% male, and the average APACHE II and SOFA scores were 19 points and 3 points. The overall ICU mortality was 17.2% without any differences between the groups. An increase in the number of infections was observed in the post-intervention group (44.5% vs. 17.9%, *p* < 0.01), especially due to an increase in the incidence of ventilator-associated pneumonia (44.6% vs. 25%, *p* < 0.001). AC decreased from 128.7 DDD in 2018 to 66.0 DDD in 2021 (rate ratio = 0.51). An increase in *Pseudomonas aeruginosa* susceptibility of 23% for Piperacillin/tazobactam and 31% for Meropenem was observed. Conclusion: The implementation of an algorithm based on rapid microbiological diagnostic techniques allowed for a significant reduction in AC and ARPs without affecting the prognosis of critically ill patients.

## 1. Introduction

Ventilator-associated pneumonia (VAP) is the most common cause of sepsis in adult ICU patients [1]. Its incidence varies between centres and countries but has increased significantly during the recent COVID-19 pandemic [2,3]. *Pseudomonas aeruginosa* is one of the major microorganisms associated with VAP [3,4,5]. Its high capacity to adapt and develop antimicrobial resistance [6] makes *P. aeruginosa* a potentially resistant microorganism when deciding on empirical antimicrobial treatments. Currently, conventional microbiological techniques take a median of 24–48 h to obtain a positive result from a quality respiratory specimen and more than 72 h to identify the microorganism and its antimicrobial susceptibility (ATB).

Rapid identification of a sepsis-causing organism is essential for early, targeted and effective antimicrobial therapy in critically ill patients [7,8]. New technologies for rapid molecular diagnosis using multiplex PCR allow rapid and accurate microbiological diagnosis using relatively simple techniques. Without adequate clinical support from multidisciplinary groups (PROA—Programme for Antimicrobial Optimisation) and the development of decision support algorithms, the implementation of these modern rapid diagnostic technologies can lead to clinician confusion, high variability in practice and suboptimal clinical impact [9,10]. Therefore, effective implementation strategies and multidisciplinary involvement are needed to ensure the correct interpretation of results and appropriate antimicrobial stewardship so that these new technologies are cost-effective rather than resource-consuming [9,10,11].

The delayed identification of the microorganism may not only delay the administration of an effective antimicrobial therapy, thereby increasing mortality, but may also lead to the overuse of empiric broad-spectrum antimicrobials, increasing healthcare costs and antimicrobial resistance [11,12].

The BIOFIRE^®^ FILMARRAY^®^ Pneumonia Panel (BPP) uses automated real-time multiplex polymerase chain reaction technology to identify nucleic acid sequences for 18 bacteria, 7 antibiotic resistance markers and 9 viruses that cause pneumonia and other lower respiratory tract infections. The BPP and other similar rapid diagnostic systems for the identification of microorganisms in respiratory specimens significantly reduce the time required to identify the microorganism [13,14,15]; however, to really improve the benefit, rapid and effective communication of results and early decision making is required.

Although there is a large amount of literature on the impact of rapid diagnostic tests for GNB infections [13,14,15], most of the data focus on measures of microbiological processes or antimicrobial administration time. In contrast, data describing the benefits on antimicrobial consumption and the impact on local resistance patterns are scarce. In this study, we evaluated the impact of using a consensus-based decision support algorithm based on rapid microbiological techniques on microbial resistance and antimicrobial use from an epidemiological perspective.

### 1.1. Primary Objective

The primary objective was to evaluate the impact of the application of a consensus-based rapid diagnostic algorithm for decision support on antimicrobial consumption and *Pseudomonas aeruginosa* bacterial resistance in all consecutive critically ill patients admitted to the ICU during the study period.

### 1.2. Primary Outcome

The primary outcome was the variation in overall antibiotic consumption measured in DDDs (defined daily doses) and in the resistance pattern of *Pseudomonas aeruginosa*.

### 1.3. Secondary Outcomes

The secondary outcomes included crude ICU mortality in the overall pre- and post-intervention population, length of stay and number of days requiring invasive mechanical ventilation.

### 1.4. Endpoints

The primary endpoint was annual antibiotic (ATB) consumption measured in defined daily doses (DDDs).

The secondary endpoints were the crude annual ICU mortality and the variation in the ATB sensitivity of *Pseudomonas aeruginosa* in the post-intervention group.

## 2. Material and Method

### 2.1. Study Design and Population

We conducted a retrospective pre-intervention/post-intervention study comparing the management of critically ill patients admitted to the 28-bed intensive care unit (ICU) before and after the implementation of a consensus-based rapid diagnostic algorithm for decision support. The pre-intervention period was January 2018 through December 2018, and the post-intervention period was January 2019 to December 2021.

In 2018, very high microbial resistance (>40%) to the antimicrobials used as the empirical treatment for respiratory infections (meropenem and piperacillin tazobactam) by *Pseudomonas aeruginosa* was evident (Figure 1).

The PROA team agreed to apply a rapid diagnostic algorithm (Figure 2) starting from January 2019 in all patients with a suspected or confirmed lower respiratory tract infection (LRTI) at risk of potential MDR *P. aeruginosa* infection (Appendix A) to reduce the possibility of the use of an inappropriate empirical antibiotic treatment. Patients admitted in 2018 were considered the reference group (pre-intervention group), and comparisons were made with patients admitted in 2019–2021 (post-intervention group).

To improve compliance with the clinical decision-making algorithm, we employed a clustered approach for different actions:(1)The clinical decision algorithm was dissemination to all ICU staff physicians from the PROA team through regular face-to-face meetings.(2)Educational lectures related to the methodology and impact of antimicrobial treatment optimisation were offered to all ICU medical staff.(3)Biofire^®^ Panel Pneumonia results were communicated in real time to the requesting physician via phone and electronic medical records.(4)Prospective audits were performed by the PROA team with real-time intervention and feedback to ICU attending physicians during the intervention period for all patients with a suspected LRTI.

### 2.2. Laboratory Methods

In both the pre- and post-intervention periods, cultures of respiratory samples (bronchial aspirate (BA) or bronchoalveolar lavage (BAL)) were performed according to the standard techniques.

Gram staining of the respiratory samples was performed in all patients to assess sample quality. Samples considered to be of poor quality were not processed and a new respiratory sample was requested.

During the intervention period, the BIOFIRE^®^ FilmArray PNEUMONIA^®^ panel was also used according to the manufacturer’s instructions following the consensus algorithm.

### 2.3. Reporting Methods

In both periods, the results of the respiratory Gram stain or identification and antibiogram were communicated by telephone to the intensivists from the microbiology service within 30 min of microorganism identification.

During the intervention period, a microbiologist communicated the BPP results directly to a member of the ICU care team within 2 h of obtaining the quality respiratory specimen. Overnight and weekend results were reported directly to the intensivist by the microbiology technician.

No restrictive antibiotic policy was applied, and the attending physician was free to choose the antimicrobial to be administered during their night duty if it was a microorganism other than Pseudomonas, in which case, the algorithm suggested starting aztreonam (AZT) at a dose of 8 g/day as a continuous perfusion. The antibiotic treatments were reviewed by the PROA team on the morning of the next working day and adjusted according to the pathogen-specific results, clinical interpretation, national guidelines and local antibiogram data.

#### Nosocomial Infection Prevention Measures

Similar nosocomial infection prevention measures were applied throughout the study (both periods) according to the established national protocols (Pneumonia Zero^®^, Bacteraemia Zero^®^ and Resistance Zero^®^) in the ENVIN-HELICS (ENVIN-HELICS (vhebron.net)) (accessed on 20 November 2023). Our ICU does not use selective digestive decontamination in patients under mechanical ventilation. No additional prevention bundles were implemented, with the exception of the rapid diagnostic algorithm for LRTIs in the post-intervention period.

### 2.4. Data Collection

#### 2.4.1. Clinical and Laboratory Data

Clinical and laboratory data were collected from the from the clinical information system (CIS, Centricity Critical Care^®^ by General Electric) by ETL (extract, transform and load) by SQL and Python. The CIS automatically incorporates data from all upstream devices every 2 min as well as laboratory values. In addition, physicians include patient-related information as an adverse event record (e.g., VAP) throughout the patient care process during the ICU stay.

#### 2.4.2. Antimicrobial Consumption Data

To measure antimicrobial consumption according to the WHO guidelines, we used the defined daily dose (DDD) methodology. DDD is the average daily maintenance dose of an antimicrobial substance used for its primary indication in adults [16]. The DDDs per 100 occupied bed-days were recorded pre- and post-intervention for all antibiotics with a special interest in those with antipseudomonal activity. DDD was calculated by considering the total number of grams of each antibiotic consumed in a hospital unit during a given period, divided by the DDD value set by the WHO for the antibiotic in question and by the number of stays in the same period of time. The consumption data were obtained from the pharmacy computer application and the number of stays was obtained from the hospital management computer system. These data are usually calculated on a quarterly and annual basis.

#### 2.4.3. Microbiological Data

The different bacterial isolates were obtained after seeding the samples using conventional quantitative methods according to the protocols of the Spanish Society of Infectious Diseases and Clinical Microbiology (SEIMC) [17]. Identification was performed using MALDI-TOF mass spectrometry (MALDI-Biotyper, Bruker Daltonics©, Bremen, Germany) and antibiotic sensitivity was analysed using the microdilution technique (MicroScan WalkAway plus, Beckman-Coulter©, Brea, CA, USA) following the recommendations of the European Committee on Antimicrobial Susceptibility Testing (EUCAST) [18].

Data on the incidence density of controlled intra-ICU infections were calculated according to national indicators from the ENVIN-HELICS registry.

### 2.5. Study Definitions

Ventilator-associated lower respiratory tract infection (vLRTI): A diagnosis of vLRTI was based on the presence of at least two of the following criteria: body temperature of more than 38.5 °C or less than 36.5 °C, leucocyte count greater than 12,000 cells per μL or less than 4000 cells per μL, and purulent bronchial aspirate (BA) or bronchoalveolar lavage (BAL).

Additionally, all episodes of infection had to have a positive microbiological isolation from the BA of at least 10^6^ colony-forming units (CFUs) per mL, or from bronchoalveolar lavage (BAL) of at least 10^4^ CFU per mL [19].

Ventilator-associated tracheobronchitis (VAT): VAT was defined with the aforementioned criteria with no radiographical signs of new pneumonia [19].

Ventilator-associated pneumonia (VAP): VAP was defined as the presence of new or progressive infiltrates on a chest radiograph [19].

Other definitions and the calculation of incidence densities of controlled infections are shown in the Appendix A.

### 2.6. Statistical Analysis

Differences between the pre-intervention and post-intervention cohorts in continuous variables were assessed using the two-sample t test or Wilcoxon rank-sum test depending on the distribution. Differences between categorical variables were assessed using the χ^2^ test or Fisher’s exact test, as appropriate.

To assess the change in incidence density of infectious complications or DDDs, we used rate ratios (RRs), also known as incidence density ratios (IDRs). The RR is a measure of association that compares the incidence of events occurring at different times.

*p* < 0.05 was considered statistically significant. The statistical analyses were performed using R.

## 3. Results

### 3.1. Overall Population

A total of 3635 patients were admitted consecutively during the study period. Of these, 987 (27.1%) were admitted in 2018 and formed the pre-intervention control group. A total of 979 (27.0%), 804 (22.1%) and 865 (23.8%) patients were admitted in 2019, 2020 and 2021, respectively, and formed the intervention group (*n* = 2648) (Appendix A).

The median age was 60 years, the sample population was 64% male, and the vast majority (95%) of patients were admitted as emergencies. The severity level showed an APACHE II score of 19 points, with a SOFA score of 3 points. Obesity (16.1%) and diabetes (12.1%) were the most frequent comorbidities observed. The mean ICU stay was 4 days with an overall crude ICU mortality of 17.2% and no difference between the periods. The complete clinical characteristics of the included patients are shown in Table 1.

A decrease in the mean age and severity of patients was observed between the pre- and post-intervention period. Statistically differences were observed in haemoglobin (Hb) concentration and serum CRP levels, but these differences are not clinically significant (Table 1 and Appendix A).

#### 3.1.1. Microbiological Findings

A significant increase in the number of infections was observed between the post- and pre-intervention periods, both when considering the total number of patients (14.3% vs. 8.4, *p* < 0.001) (Table 1) or the total number of controlled intra-ICU infections, with an increase from 17.9% in 2018 (pre-intervention) to 44.5% (*p* < 0.01) in 2021 (post-intervention) (Table 1). This increase was mainly due to the higher number of ventilator-associated pneumonia (VAP) cases, which increased from 25% in 2018 to 44.6% (*p* < 0.001) in 2021. A similar pattern was observed when comparing the incidence density of recorded infections. The rate ratio of episodes per 1000 mechanical ventilation days was 1.3 when comparing the pre- and post-intervention periods and reached 2.3 when comparing 2018 to 2021 (Table 2).

The 2020–2021 period corresponds to the COVID-19 pandemic, in which there was a higher number of patients requiring invasive mechanical ventilation (Table 1). However, the proportion of medical and surgical patients was similar in the pre- and post-intervention periods (Appendix A).

Methicillin-sensitive *Staphylococcus aureus* (MSSA, 15.7%), *Escherichia coli* (*E. coli*, 12.7%), *Klebsiella pneumoniae* (KP, 9.8%) and *Pseudomonas aeruginosa* (PA, 8.8%) were the four most frequently isolated microorganisms in the pre-intervention period. In the post-intervention period, *Pseudomonas aeruginosa* (15.6%, *p* = 0.04) was the most frequently isolated followed by MSSA (14.0%), *K. pneumoniae* (11.6%) and *E. coli* (9.8%) (Appendix A). Despite this increase in *P. aeruginosa* isolation, extensive drug-resistant (XDR) strains appeared less frequently in the post-intervention period (*n* = 3) compared to the pre-intervention period (*n* = 5) (data not shown).

#### 3.1.2. Antibiotic Consumption

The rapid diagnostic algorithm was applied in 354 of 2648 patients (13.4%) with a suspected LRTI: 70 (19.7%) in 2019, 135 (38.2%) in 2020 and 149 (42.1%) in 2021. The median time from sample collection to FilmArrays result was 1.48 (1.36–3.45) hours without a difference between periods.

LRTI (VAP + VAT) was diagnosed in 15 patients (21.4%) in 2019, 54 (40%) in 2020 and 105 (70.5%) in 2021. However, *P. aeruginosa* was only isolated in 4 (20.0%), 9 (16.0%) and 24 (18.2%) patients, respectively (Appendix A).

Despite the increase in the number of controlled infections, ATB consumption decreased from 128.7 DDD in 2018 to 66.0 DDD in 2021 (rate ratio = 0.51) (Figure 3A). A marked reduction in the use of meropenem (rate ratio = 0.73), piperacillin/tazobactam (rate ratio = 0.39) and ceftazidime (rate ratio = 0.27) was observed. In contrast and as expected, the use of aztreonam increased markedly (rate ratio = 66.5) (Figure 3B and Appendix A). Despite the reduction in the use of ATB in the post-intervention period, no difference was observed in crude mortality compared to the pre-intervention period (Table 1).

#### 3.1.3. *Pseudomonas aeruginosa* Susceptibility Pattern

An elevated *P. aeruginosa* resistance pattern was observed in 2018 for almost all antipseudomonal antibiotics (Figure 1), particularly for meropenem (41%) and piperacillin/tazobactam (39%), antimicrobials used in the empirical treatment of VAP, which presented higher resistances (close to 50%) in PA isolated from respiratory specimens (Figure 2). A significant increase in sensitivity to all antibiotics was observed after implementing the rapid diagnostic algorithm. Specifically, an improvement of 23% for PTZ and 31% for MRP were observed (Figure 1 and Figure 2). The increase in the use of AZT did not affect the sensitivity pattern of *P. aeruginosa*.

## 4. Discussion

While the data presented for the syndromic molecular test for nosocomial pneumonia clearly demonstrate high accuracy and the detection of many more pathogens than culture [8,9,10,11,12,13], there is still little published information demonstrating that this translates into improved antibiotic use or a clinical benefit. Thus, we conducted an epidemiological study to evaluate the impact of the implementation of a decision support algorithm on antibiotic consumption and microbial resistance patterns not only in patients with an LRTI, but in the whole population admitted to the ICU.

Our main finding was that despite an observed increase in the incidence of nosocomial infections, the implementation of a decision support algorithm based on rapid diagnostic techniques was associated with lower antibiotic consumption and an increase in antimicrobial susceptibility in the ICU. Furthermore, this reduction in antibiotic use was not associated with an increase in crude ICU mortality. This suggests that the algorithm led to a reduction in antibiotic overuse.

The development of antimicrobial resistance is a normal evolutionary process for microorganisms, but it is accelerated by the selective pressure exerted by the widespread use of antimicrobials [20]. There is a strong association between antimicrobial resistance and antimicrobial use levels, implying that a reduction in unnecessary antimicrobial consumption could favourably affect resistance [21,22].

Several risk factors expose critically ill patients to an increased risk of colonization and infection by multidrug-resistant organisms, such as treatment with immunosuppressive drugs, use of invasive devices, exposure to a wide range of antibiotics and prolonged hospitalizations [23].

Our results support a remarkable increase in LRTIs in the post-intervention period. Most of this period (2020–2021) includes the COVID-19 pandemic, which resulted in an uncontrolled influx of critically ill patients, often receiving unnecessary antibiotic therapy [24,25]. A Centers for Disease Control (CDC) report published in February 2021 describes outbreaks of antimicrobial-resistant infections in COVID-19 units [26], with a marked increase in nosocomial infections, most of which were caused by multidrug-resistant organisms [27].

We did not observe an increase in XDR strains and overall susceptibility to *P. aeruginosa* improved over the years. These findings agree with those of Langford BJ et al. [28], who reported no association between COVID-19 and the incidence of resistant *P. aeruginosa* (IRR 1.10, 95% CI: 0.91–1.30), nor with the proportion of resistant cases (RR 1.02, 95% CI: 0.85–1.23).

Despite this increase in the nosocomial infection, the application of a decision support algorithm based on rapid microbiological diagnostic techniques decreased the consumption of antibiotics and increased the microorganisms’ sensitivity to old antimicrobials.

Different authors [8,13,29,30,31] have reported that multiplex bacterial PCR testing of quality respiratory samples decreases the duration of inadequate antibiotic treatment of patients admitted to the hospital with pneumonia and risk of Gram-negative bacilli infection. However, most of these studies have been performed in general hospitalization patients [8,29,31], or in haematologic [30] or paediatric patient groups [13] and little information exists on the impact of these techniques in critically ill adult patients.

To the best of our knowledge, only one study has included critically ill patients, and its findings agree with our results. Specifically, Rizk NA et al. [32] reported a decrease in resistance rates among *Acinetobacter baumannii* to imipenem from 81% in 2018 to 63% in 2020 with the implementation of antibiotic stewardship and an infection control policy, especially in ICUs, with a decrease in carbapenem use at the hospital level. In addition, an open label, randomized, parallel, multicentre study (INHALE WP3) [33] has been designed to explore the potential impact of rapid molecular diagnostics coupled with a prescribing algorithm, with the goal of achieving non-inferiority in clinical cure of pneumonia and superiority in terms of antimicrobial stewardship, compared with the standard care. We hope that this study, suspended during the COVID-19 pandemic, can provide valuable information on an unmet demand for intensivists.

LTRIs associated with mechanical ventilation represent the most frequent infectious episodes in patients admitted to the intensive care unit (ICU) requiring mechanical ventilation [1]. LRTIs are associated with a high mortality rate (more than 50%) and a significant impact on ICU length of stay, antibiotic use and overall healthcare costs [1,4,5]. As we observed in our study, Gram-negative pathogens are responsible for the majority of VAP cases, especially non-fermenting Gram-negative pathogens like *Pseudomonas aeruginosa.* This has also been reported by other authors before [19] and during the pandemic period [34]. However, a recent meta-analysis that included a study period similar to ours (2019–2021) reported that *Pseudomonas aeruginosa* (*n* = 65) was not the first, but the third most commonly isolated Gram-negative MDR organism after *Klebsiella pneumoniae* (*n* = 169) and *Acinetobacter baumannii* (*n* = 148) [35]. The study population, the burden of COVID-19, the burden of non-COVID-19 respiratory infections, local epidemiology and especially antimicrobial prescribing practices may partially explain the difference between the overall data and our findings.

Our study has several limitations that we must acknowledge. First, we did not design our protocol to assess the impact of individual interventions on outcomes. All patients for whom the algorithm was applied received an ATB after obtaining microbiological samples and we did not expect an improvement in administration times. Thus, our approach was epidemiological with the aim of studying the impact of the algorithm on “macro” indicators such as the annual consumption of ATBs or the variation in sensitivity over the years.

Second, the study had a retrospective, nonrandomized design. Given the before/after design of the study, the results could be biased due to residual confounding factors that were not considered. However, the study was designed to address a clinical need and represents real-life data after applying a decision support algorithm.

Third, our study was conducted at a single centre, and the clinical results may not be directly translatable to other centres. It is necessary to consider that the findings may be influenced by the appropriate application and high acceptance of the decision algorithm.

Fourth, we only presented the variation in the sensitivity pattern for *Pseudomonas aeruginosa* because it was the only microorganism that showed a high level of resistance during the pre-implementation period.

Fifth, we cannot rule out that the implementation of an action plan may have affected the final result, which was unrelated to the technique used. However, this action plan was based not only on the application of an antimicrobial treatment optimisation protocol using a decision algorithm with rapid techniques but also on the field work of the PROA team. Although the activity of the PROA team is of fundamental importance, the early availability of microbiological data greatly enhanced its impact.

Finally, we did not record other outcomes such as the duration or adequacy of antimicrobial treatment. The aim of our study was to assess general indicators that reflect the adequacy of the overall treatment of critically ill patients during the study period.

## 5. Conclusions

The implementation of a decision support algorithm based on rapid microbiological diagnostic techniques resulted in a marked reduction in antibiotic consumption and bacterial resistance without affecting the prognosis of critically ill patients. The PROA team is essential for the development and implementation of these decision support algorithms.

## Figures and Tables

**Figure 1 biomedicines-11-03330-f001:**
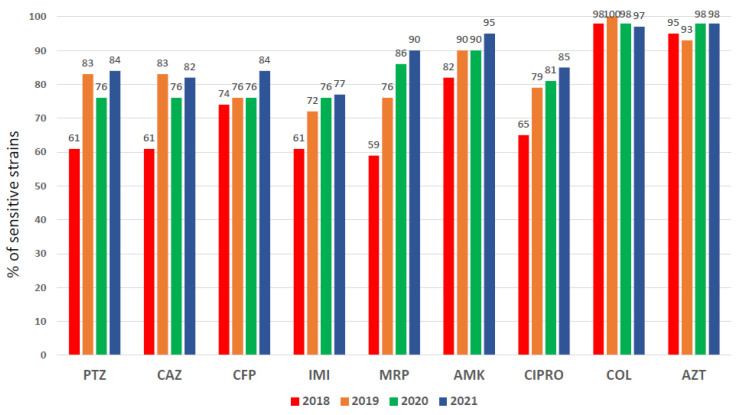
Resistance pattern of Pseudomonas aeruginosa strains for the main antimicrobials in the pre- (2018, *n* = 987) and post-intervention periods (2019, *n* = 979; 2020, *n* = 804; and 2021, *n* = 865).

**Figure 2 biomedicines-11-03330-f002:**
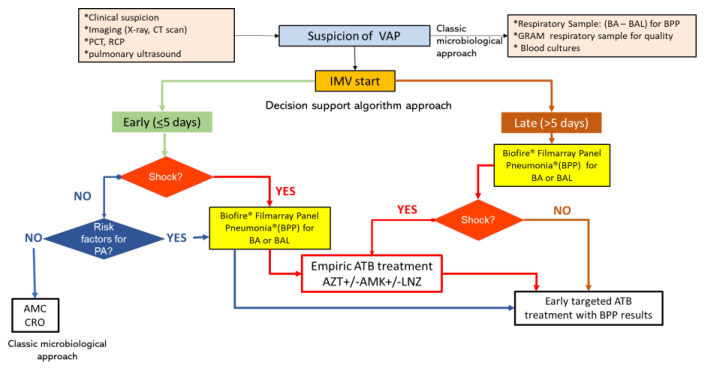
Consensus decision support algorithm based on rapid microbiological diagnostic techniques. The algorithm is based on 3 key points: (1) the clinical diagnosis of probable VAP including PCT and CRP; (2) the patient’s own conditions and here, MV time is important to determine early or late VAP and associated microorganisms; and (3) FilmArrays are requested in all patients at risk of MDR bacteria infection in order to be able to adjust the treatment early. PCT: procalcitonin, RCP: reactive C protein, VAP: ventilator-associated pneumonia, BA: bronchial aspirate, BAL: bronchoalveolar lavage, IMV: invasive mechanical ventilation, AMC: amoxicillin/clavulanate, CRO: ceftriaxone, AZT: aztreonam, AMK: amikacin, LNZ: linezolid, ATB: antibiotic.

**Figure 3 biomedicines-11-03330-f003:**
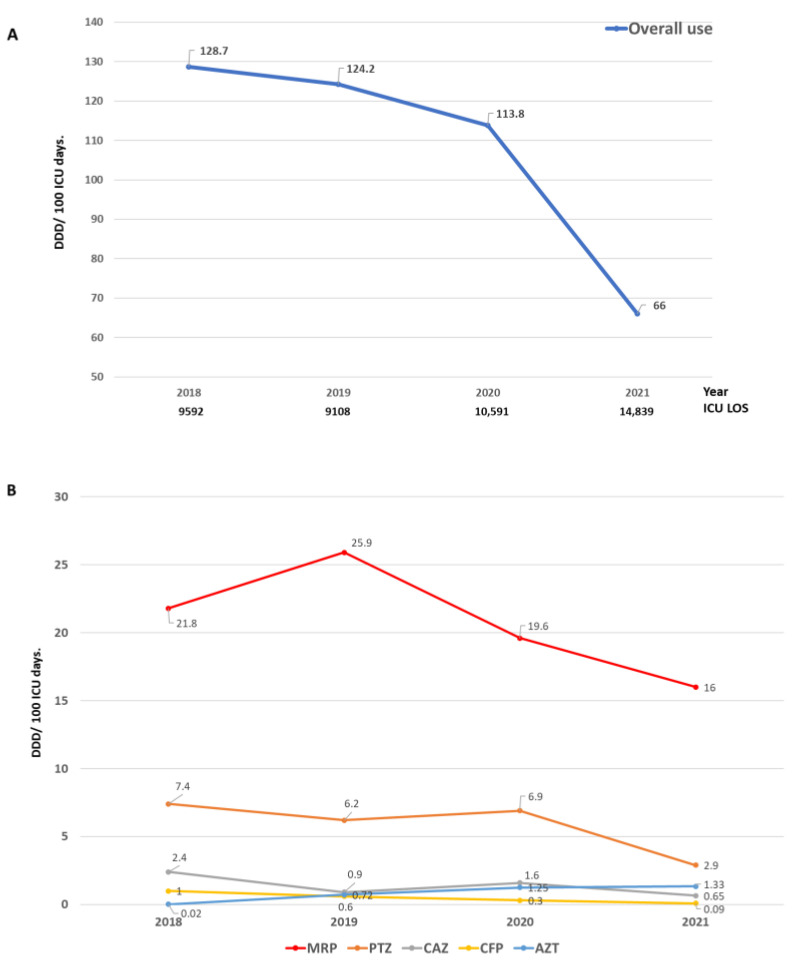
Antibiotic consumption expressed in defined daily doses (DDDs) in the years included in the study. (**A**) Overall antibiotic consumption and (**B**) consumption of the main controlled antibiotics (MRP: meropenem, PTZ: piperacillin/tazobactam, CAZ: ceftazidime, CFP: cefepime, AZT: aztreonam).

**Table 1 biomedicines-11-03330-t001:** Characteristics of the 3635 critically ill patients included in the study according to the study period and according to the years considered within each period.

Study Period	Overall	Pre-Intervention	Intervention			*p*-Value
Variable	*n* = 3635	2018 (*n* = 987)	2019 (*n* = 979)	2020 (*n* = 804)	2021 (*n* = 865)	
Demographics and Severity						
Age, mean (Q1–Q3)	60 (50–72)	64 (52–73)	64 (50–73)	63 (50–72)	61 (49–72) ***	0.009
Male, *n* (%)	2350 (64.6)	634 (64.2)	620 (63.3)	536 (66.7)	560 (64.7)	0.52
APACHE, mean (Q1–Q3)	19 (14–25)	20 (15–25)	20 (15–25)	18.5 (14–24) ***	19 (14–25)	0.003
SOFA, mean (Q1–Q3)	3 (1–6)	3.0 (2.0–5.1)	2.2 (1.0–5.0) ***	3.0 (2.0–6.0)	3 (2.0–6.0)	<0.001
Patients Type						
Surgical, *n* (%)	955 (26.3)	275 (27.9)	300 (30.7) ***	199 (24.7)	181 (21.0) **	<0.001
Medical, *n* (%)	2680 (73.7)	712 (72.1)	679 (69.3)	605 (75.3)	684 (79.0) **	<0.001
COVID-19, *n* (%) within medical patients	398 (14.8)	0 (0.0)	0 (0.0)	173 (28.5)	225 (32.9)	NA
Comorbidities						
Obesity, *n* (%)	569 (15.6)	123 (12.4)	146 (14.9) **	135 (16.8) ***	165 (19.0) ***	<0.001
Diabetes, *n* (%)	825 (22.7)	239 (24.2)	229 (23.4)	174 (21.6)	183 (21.1)	0.35
Chronic heart disease, *n* (%)	161 (4.4)	61 (6.2)	35 (3.6) **	30 (3.7) *	35 (4.0)	0.01
COPD, *n* (%)	400 (11.0)	126 (12.7)	120 (12.2)	89 (11.0)	65 (7.5) ***	0.001
Chronic Rennal failure, *n* (%)	339 (9.3)	93 (9.4)	88 (9.0)	78 (9.7)	80 (9.2)	0.96
Immunosupression, *n* (%)	159 (4.4)	43 (4.3)	33 (3.4)	47 (5.8)	36 (4.1)	0.29
Laboratory						
Hemoglobin g/dL, median (Q1–Q3)	10.0 (8.5–12.0)	10.3 (8.6–12.2)	10.1(8.6–12.0)	9.7 (8.4–11.5) ***	10.1 (8.6–12.1)	<0.001
WBC count 10^3^/uL, median (Q1–Q3)	10.4 (8.0–13.6)	10.8 (8.1–13.9)	10.3 (7.9–13.5)	10.4 (8.3–13.5)	10.7 (8.2–13.8)	0.21
Serum creatinine mg/dL, median (Q1–Q3)	0.7 (0.5–1.1)	0.7 (0.6–1.1)	0.7 (0.5–1.1)	0.7 (0.5–1.1)	0.7 (0.5–1.1)	0.19
PCT ng/mL, median (Q1–Q3)	2.25(0.57–7.27)	3.17 (1.27–8.65)	2.65(0.94–8.34)	1.60(0.34–6.0)	1.11(0.26–5.23)	<0.001
RCP mg/dL, median (Q1–Q3)	9.7 (4.2–18.9)	9.9 (5.3–18)	9.5 (4.9–16.4)	9.4 (4.8–17.0)	8.5 (4.2–16.0) ***	0.01
Microbiologically Confirmed Infections During ICU Stay						
Total number of infections, *n* (%)	463 (100)	83 (17.9)	55 (11.9)	119 (25.7)	206 (44.5)	<0.001
Ventilator-associated pneumonia (VAP), *n* (%)	163 (35.2)	21 (25.3)	12 (21.8)	38 (32.0)	92 (44.6) ***	0.01
Bacteraemia secondary to other septic foci (BS), *n* (%)	57 (12.3)	17 (20.5)	9 (16.6)	14 (11.8)	17 (8.3) **	0.02
Bacteraemia of unknown origin (BUNK), *n* (%)	69 (14.9)	9 (10.8)	12 (21.8)	19 (16.0)	29 (14.0)	0.33
Catheter-associated urinary tract infection (CAUTI), *n* (%)	50 (10.8)	8 (9.6)	3 (5.4)	10 (8.4)	29 (14.0)	0.19
Ventilator-associated tracheobronchitis (VAT), *n* (%)	37 (8.0)	5 (6.0)	3 (5.4)	16 (13.4)	13 (6.4)	0.08
Catheter-related bacteraemia (CRB), *n* (%)	51 (11.0)	4 (4.9)	3 (5.4)	22 (18.4) **	22 (10.7)	0.008
Intra-abdominal infections (IAI), *n* (%)	10 (2.1)	4 (4.9)	5 (9.1)	0 (0%)	1 (0.5) *	<0.001
Skin and soft tissue infection (SSTI), *n* (%)	9 (2.0)	4 (4.8)	3 (5.4)	0 (0%)	2 (1.0) *	0.01
Others, *n* (%)	17 (3.7)	11 (13.2)	5 (9.1)	0 (0%)	1 (0.5) ***	<0.001
Main Micro-Organisms Isolated During ICU Stay						
Total number of microorganisms isolated, *n* (%)	602 (100)	102 (17.0)	75 (12.4)	159 (26.4)	266 (44.2)	<0.01
*Staphylococcus aureus*	86 (14.4)	16 (15.7)	9 (12.0)	23 (14.5)	38 (14.2)	0.85
*Escherichia coli*	62 (10.4)	13 (12.7)	7 (9.3)	12 (7.5)	30 (11.3)	0.50
*Klebsiella pneumoniae*	68 (11.3)	10 (9.8)	10 (13.3)	18 (11.3)	30 (11.3)	0.91
*Pseudomonas aeruginosa*	87 (14.4)	9 (8.8)	9 (12.0)	20 (12.6)	49 (18.4) *	0.07
*Enterobacter aerogenes*	19 (3.1)	7 (6.8)	0 (0%)	5 (3.1)	7 (2.6)	0.06
*Serratia marcescens*	26 (4.3)	6 (5.8)	1 (1.3)	5 (3.1)	14 (5.3)	0.34
*Haemophilus influenzae*	28 (4.6)	5 (4.9)	4 (5.3)	8 (5.0)	11 (4.1)	0.94
*Enterococcus faecium*	13 (2.1)	4 (3.9)	5 (6.7)	1 (0.6)	3 (1.1)	0.79
*Klebsiella oxytoca*	15 (2.5)	4 (3.9)	1 (1.3)	4 (2.5)	6 (2.6)	0.72
*Proteus mirabilis*	11 (1.8)	3 (2.9)	0 (0%)	5 (3.1)	3 (1.1)	0.22
*Citrobacter* spp.	16 (2.6)	3 (2.9)	0 (0%)	4 (2.5)	9 (3.4)	0.45
*Enterobacter cloacae*	32 (5.3)	3 (2.9)	7 (9.3)	10 (6.3)	12 (4.5)	0.24
*Enterococcus faecalis*	33 (5.5)	3 (2.9)	2 (2.7)	18 (11.3) *	10 (3.7)	0.02
Others	106 (17.7)	16 (15.6)	20 (26.6)	26 (16.3)	44 (16.4)	0.18
Complications and Outcome						
Invasive Mechanical ventilation, *n* (%)	1802 (49.6)	425 (43.1)	421 (43.0)	476 (59.2) ***	480 (55.5) ***	<0.001
LOS ICU, mean (Q1–Q3)	4.1(2.0–10.2)	4.0 (2.0–8.0)	3.6 (1.8–7.7) **	4.8 (2.2–14.0) ***	5.4(2.2–14.1) ***	<0.001
Crude ICU Mortality, *n* (%)	625 (17.2)	165 (16.7)	148 (15.1)	158 (19.7)	154 (17.8)	0.08

* *p* < 0.05; ** *p* < 0.01; *** *p* < 0.001 for bivariate comparison between 2018 year (pre-intervention) as reference group and years of post-intervention period.

**Table 2 biomedicines-11-03330-t002:** Incidence density of controlled infections during ICU stay according to pre- or post-intervention period (**A**) and differentiating years within the post-intervention period (**B**).

(A)							
Study Period	Pre-Intervention	Intervention			RR		95%ICRR
Variable	2018 (*n* = 987) (1)	2019–21 (*n* = 2648) (2)			2 vs. 1		
Incidence density of reported ICU-associated infections							
VAP episodes/1000 mechanical ventilation days	5.5	7.33			1.33		0.4–4.1
CAUTI episodes/1000 urinary catheter days	1.30	1.55			1.19		0.1–11.2
CRB and BUNK episodes/1000 catheter days	1.7	2.8			1.64		0.2–11.0
BS episodes/1000 ICU days	2.3	1.3			0.56		0.1–4.8
**(B)**							
	**Pre-Intervention**	**Intervention**			**RR (95% CI)**		
Variable	2018 (1) (*n*= 987)	2019 (2) (*n* = 979)	2020 (3) (*n* = 804)	2021 (4) (*n* = 865)	RR 2 vs. 1 (95% CI)	RR 3 vs. 1 (95% CI)	RR 4 vs. 1 (95% CI)
VAP episodes/1000 mechanical ventilation days	5.5	2.82	6.28	12.9	0.5 (0.3–1.4)	1.14 (0.7–1.9)	2.3 ** (1.4–3.7)
CAUTI episodes/1000 urinary catheter days	1.30	0.46	1.18	3.01	0.35 (0.1–1.3)	0.90 (0.3–2.2)	2.3 ** (1.1–6.1)
CRB and BUNK episodes/1000 catheter days	1.7	1.8	3.0	3.5	1.05 (0.4–2.5)	1.8 (0.5–2.0)	2.0 (1.0–3.5)
BS episodes/1000 ICU days	2.3	1.1	1.4	1.4	0.5 (0.2–1.2)	0.6 (0.2–1.7)	0.6 (0.3–1.3)

RR = Rate Ratio or Incidence Density Ratio. 95%ICRR: 95% confidence interval of Rate Ratio intervention vs pre intervention periods. Compares the incidence of events occurring at different times. VAP = Ventilator-associated pneumonia, CAUTI = Catheter-associated urinary tract infection, BRC: catheter-related bacteraemia, BS= Bacteraemia secondary to other septic foci, BUNK = Bacteraemia of unknown origin. ICU = Intensive care unit. ** *p* value <= 0.05.

## Data Availability

The data supporting the conclusions of this study are available from Hospital Universitario Joan XXIII, but restrictions apply to the availability of these data, which were used under Hospital Joan XXIII authorization for the present study and are therefore not publicly available. However, the data can be obtained from the corresponding author (RA) upon reasonable request and with the permission of Hospital Joan XXIII.

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
