# Peer review of "Clinical and Microbiological Impact of Implementing a Decision Support Algorithm through Microbiologic Rapid Diagnosis in Critically Ill Patients: An Epidemiological Retrospective Pre-/Post-Intervention Study"

_biomedicines, 2023, doi:10.3390/biomedicines11123330_

Round 1

Reviewer 1 Report

Comments and Suggestions for Authors

A very interesting and valuable study has been carried out. The process and results of the study are quite well described. However, in my opinion, important data is missing.

Methods section.

• Incomplete decision algorithm (Figure 2). It includes the BIOFIRE® FILMARRAY® Pneumonia Panel (BPP) method in two places. However, the method itself does not determine the decision. Decision-making is determined by the result. Therefore, this algorithm does not give the impression that decisions were made based on the result. A more detailed explanation is necessary.

• The same algorithm also mentions "Respiratory sample". It is unclear whether there was only microscopy or microscopy and culture. Also, the hierarchy of sample results for decision-making is not clear. Please clarify it.

Results section.

• Correlation between detected pathogens and antibiotic susceptibility between the BIOFIRE® FILMARRAY® Pneumonia Panel (BPP) and conventional bacteriological methods was not reported.

• It is not stated how quickly the test results were obtained in daily practice.

• There is no information about Acinetobacter, one of the most common nosocomial pathogens.

• It is not clear what part BIOFIRE® FILMARRAY® Pneumonia Panel (BPP) had for the positive changes, and what part - conventional bacteriological methods. Or simply the implementation of a standard action plan per se?

Author Response

Reviewer 1

Dear reviewer, thank you very much for your constructive comments. We have incorporated your suggestion into the new manuscript and feel that it is now more robust and clearer for readers. Below we respond point by point to each comment.

Methods section.

1.- Incomplete decision algorithm (Figure 2). It includes the BIOFIRE® FILMARRAY® Pneumonia Panel (BPP) method in two places. However, the method itself does not determine the decision. The decision is determined by the outcome. Therefore, this algorithm does not give the impression that decisions have been made based on the outcome. A more detailed explanation is needed.

REPLY: Thank you for your comment. We regret that we failed to explain the algorithm adequately. We have modified the algorithm to try to emphasise that the decision is based on the results of the technique, especially when the patient is not in shock. In addition, we have added a brief explanation in the figure 2. We hope that the algorithm can now be properly interpreted.

2.- The same algorithm also mentions "respiratory sample". It is not clear whether there was only microscopy or microscopy and culture. Also, the hierarchy of sample results at the time is not clear. Please clarify.

REPLY: Thank you for your comments. We have added that the respiratory samples (BA and BAL) are for filmArrays and the GRAM is to assess the quality of the sample to be processed. We hope that now the algorithm is clearer for the readers.

Results section.

1.- Correlation between pathogens detected and antibiotic susceptibility between the BIOFIRE® FILMARRAY® Pneumonia Panel (BPP) and conventional bacteriological methods was not reported.

REPLY: The reviewer is right; this analysis was not performed because it was not the objective of the present study. Several studies have already been published showing a good correlation between Filmarrays and cultures, so our objective was to assess the epidemiological impact of the application of an algorithm based on rapid tests on the entire ICU and not only on the patients in whom this technique was requested. 

2.- lt is not stated how quickly the test results were obtained in daily practice.

REPLY: Thank you for your comment. The reviewer is right and this is an important piece of information. The median time from sample collection to FilmArrays result was 1.48 (1.36-3.45) hours, with no differences between the periods. We add this information in section 3.1.2. Antibiotic Consumption

3.- There is no information on Acinetobacter, one of the most common nosocomial pathogens.

REPLY: Thank you for your comment. The reviewer is right, and Acinetobacter is a common pathogen in many ICUs in our country and worldwide. However, as with MRSA, our ICU has almost no Acinetobacter infections. As can be seen in e-Table 3, only 5 Acinetobacter isolates have been obtained in patients with VAP over 3 years. 

e-Table 3: Microorganisms isolated in patients with ventilator-associated pneumonia according to the study period. 

Pre-intervention

Intervention

Year

2018

2019

2020

2021

 Nº Patients/ Nº Microorganisms

21 /27

12/20

38/56

92/132

Staphylococcus aureus n (%)

9 (33.3)

4 (20.0)

15 (26.8)

28 (21.0)

Klebsiella spp n (%)

4 (14.8)

----

7 (12.5)

16 (12.1)

Escherichia coli n (%)

3 (11.1)

3 (15.0)

3 (5.3)

12 (9.0)

Pseudomonas aeruginosa n (%)

3 (11.1)

4 (20.0)

9 (16.0)

24 (18.2)

Haemophilus influenzae n (%)

2 (7.4)

3 (15.0)

3 (5.3)

6 (4.5)

Proteus mirabilis n (%)

1 (3.7)

----

2 (3.5)

2 (1.5)

Stenotrophomona maltophilia  n (%)

1 (3.7)

----

1 (1.8)

2 (1.5)

Streptococcus pneumoniae n (%)

1 (3.7)

2 (10.0)

1 (1.8)

4 (3.0)

Enterobacter spp. n (%)

1 (3.7)

----

2 (3.6)

12 (9.0)

Acinetobacter spp. n (%)

----

1 (5.0)

1 (1.8)

3 (2.3)

Staphylococcus aureus methicillin resistant n (%)

----

1 (5.0)

----

2 (1.5)

Serratia marcescens

----

----

3 (5.3)

6 (4.5)

Aspergillus spp

----

----

----

7 (5.3)

Others

2 (7.4)

2 (10.0)

9 (16.0)

8 (6.0)

4.- It is not clear what role BIOFIRE® FILMARRAY® Pneumonia Panel (BPP) played in the positive changes, and what role conventional bacteriological methods played, or simply the application of a standard action plan per se?

REPLY: Thank you very much for the comment, which is very appropriate and difficult to answer. We have tried to describe this in the discussion because during the intervention period, there was a significant increase in intra-ICU infection and especially VAP. Despite this, antibiotic consumption decreased. On the other hand, during the whole study period, classical microbiological techniques did not change. We recognise that implementing an improvement or control action can impact per se on the final outcome. However, we cannot demonstrate this. Given these considerations, we have added a new limitation related to your comment, which is very appropriate.

Reviewer 2 Report

Comments and Suggestions for Authors

Comments to the manuscript intitled "Clinical and Microbiological Impact of Implementing a Decision Support Algorithm Through Microbiologic Rapid
Diagnosis in Critically Ill Patients. An Epidemiological Retrospective Pre/Post-Intervention Study" by Alejandro Rodríguez et al.

Minor comments:

 1 -In table 1 PCT was not reported between the laboratory data, please correct or show the results and how many time PCT was determined in controls and in post intervention subjects.

2 - Please enter in figure 2 on which biological sample to run the BIOFIRE® FILMARRAY® pneumonia panel. I think you intend to test the sputum or BAL or the positive blood cultures.

Major comments:

1 - Based on what has been reported and stated by you that PCR and PCT have no clinical relevance, why do you recommend determining it?

2 - As is recognized by many, the PCR is a very non-specific test but with a high negative predictive value. Furthermore, the PCT is recommended to verify the effectiveness of the therapy if it is high. Why don't you recommend determining them after the IMV in the algorithm?

Author Response

Reviewer 2

We appreciate the reviewer's comments and the possibility to improve our manuscript based on them.  We respond point by point to the comments below. We believe that the manuscript is now clearer for readers.

Minor Comments

1 -In table 1 PCT was not reported between the laboratory data, please corrector show the

results and how many time PCT was determined in controls and in post intervention subjects.

REPLY: The reviewer is right. We apologise for this error. The PCT values have been included in the new Table 1.

2 - Please enter in figure 2 on which biological sample to run the BIOFIRE® FILMARRAY®

pneumonia panel. 1 think you intend to test the sputum or BAL or the positiva blood cultures.

REPLYr: Thank you for your comment. We have added that filmarrays should be performed on BA (bronchoaspirate) or BAL (bronchoalveolar lavage). We do not mention sputum although this technique is validated for this type of sample, we consider that it is not a sample to be obtained in intubated patients at risk of VAP.

Major Comments

1.-Based on what has been reported and stated by you that CRP and PCT have no clinical relevance, why do you recommend determining it?

Reply: We did not perform an assessment of the clinical relevance of PCT or CRP in our study. These biomarkers with their advantages and limitations are included in the first part of the algorithm where we try to obtain as much clinical, laboratory, biomarker, lung ultrasound, X-ray etc. information as possible to allow a more adequate pre-test diagnosis to request rapid diagnostic techniques.

2.- As many recognise, PCR is a very non-specific test but with a high negative predictive value.

negative. In addition, PCT is recommended to verify the efficacy of therapy if it is elevated.

Why is it not recommended in the algorithm to be determined after IMV?

REPLY: We appreciate the reviewer's comment and it is clear that we failed to adequately communicate the description of the algorithm.

Our algorithm starts with the clinical diagnosis of VAP in a ventilated patient. Here, with the patient already ventilated, is when PCR and PCT are requested to complete and improve the clinical diagnosis.

In other words, our algorithm is based on 3 key points. 1) the clinical diagnosis of probable VAP where PCT and PCR are included, 2) the patient's own conditions and here the MV time is important to determine early or late VAP and the associated microorganisms. However, this is one more piece of information when we have already made the diagnosis to determine the risk of MDR in our patients.  Finally (3) FilmArrays are requested in all patients at risk of MDR in order to be able to adjust treatment early.  We have improved the explanation of the algorithm.

Round 2

Reviewer 1 Report

Comments and Suggestions for Authors

None

Reviewer 2 Report

Comments and Suggestions for Authors

no comments.